# Impact of Treatment with RUTF on Plasma Lipid Profiles of Severely Malnourished Pakistani Children

**DOI:** 10.3390/nu12072163

**Published:** 2020-07-21

**Authors:** Engy Shokry, Kamran Sadiq, Sajid Soofi, Atif Habib, Naveed Bhutto, Arjumand Rizvi, Imran Ahmad, Hans Demmelmair, Olaf Uhl, Zulfiqar A. Bhutta, Berthold Koletzko

**Affiliations:** 1Department of Pediatrics, Ludwig-Maximilians-University Paediatrics, Division of Metabolic and Nutritional Medicine, Dr. von Hauner Children’s Hospital, 80337 Munich, Germany; engy.shokry@med.uni-muenchen.de (E.S.); Hans.Demmelmair@med.uni-muenchen.de (H.D.); Olaf.Uhl@med.uni-muenchen.de (O.U.); 2Department of Pediatrics & Child Health, The Aga Khan University, Stadium Road, P.O. Box 3500, Karachi 74800, Pakistan; kamran.sadiq@aku.edu (K.S.); sajid.soofi@aku.edu (S.S.); 3Center of Excellence in Women & Child Health, The Aga Khan University, Stadium Road, P.O. Box 3500, Karachi 74800, Pakistan; atif.habib@aku.edu (A.H.); naveed.bhutto@aku.edu (N.B.); arjumand.rizvi@aku.edu (A.R.); imran.ahmad@aku.edu (I.A.); 4Centre for Global Child Health, The Hospital for Sick Children, Toronto, ON M5G 0A4, Canada

**Keywords:** severe acute malnutrition (SAM), metabolomics, glycerophospholipids, essential fatty acids (EFA), lysophosphatidylcholines (LysoPC)

## Abstract

(1) Background: Little is known on impacts of ready-to-use therapeutic food (RUTF) treatment on lipid metabolism in children with severe acute malnutrition (SAM). (2) Methods: We analyzed glycerophospholipid fatty acids (FA) and polar lipids in plasma of 41 Pakistani children with SAM before and after 3 months of RUTF treatment using gas chromatography and flow-injection analysis tandem mass spectrometry, respectively. Statistical analysis was performed using univariate, multivariate tests and evaluated for the impact of age, sex, breastfeeding status, hemoglobin, and anthropometry. (3) Results: Essential fatty acid (EFA) depletion at baseline was corrected by RUTF treatment which increased EFA. In addition, long-chain polyunsaturated fatty acids (LC-PUFA) and the ratio of arachidonic acid (AA)/linoleic acid increased reflecting greater EFA conversion to LC-PUFA, whereas Mead acid/AA decreased. Among phospholipids, lysophosphatidylcholines (lyso.PC) were most impacted by treatment; in particular, saturated lyso.PC decreased. Higher child age and breastfeeding were associated with great decrease in total saturated FA (ΣSFA) and lesser decrease in monounsaturated FA and total phosphatidylcholines (ΣPC). Conclusions: RUTF treatment improves EFA deficiency in SAM, appears to enhance EFA conversion to biologically active LC-PUFA, and reduces lipolysis reflected in decreased ΣSFA and saturated lyso.PC. Child age and breastfeeding modify treatment-induced changes in ΣSFA and ΣPC.

## 1. Introduction

Globally, around 73 million children under the age of 5 years are malnourished [1]. Malnutrition is an important risk factor for serious short- and long-term adverse effects on health including stunting, increased morbidity and mortality from infectious and other disorders, and altered cognitive performance [2]. Children with severe acute malnutrition (SAM) suffer from muscle wasting and reduced height-for-age (HAZ), weight-for-age (WAZ), and weight-for-height (WHZ) z-scores below −3 SD [3]. The pathophysiology of SAM is not fully understood, but is associated with severe alterations of protein, glucose, and lipid metabolism, including depletion of essential polyunsaturated fatty acids (PUFA) [4,5,6,7].

Ready-to-use therapeutic foods (RUTF) are increasingly used for prevention and treatment of SAM and were reported to help reduce morbidity and mortality [8]. Energy and micronutrient rich paste have been used as RUTF and replaced therapeutic milks in the treatment of children with SAM [9,10]. Information on how RUTF influences the metabolic response and especially lipid metabolism is still limited [11].

Most of the previous studies on malnutrition focused on the metabolic or lipid changes observed in malnourished in comparison to heathy children rather than those in response to nutritional intervention in malnourished children [11,12]. Also, lipidomics methods applied in previous studies typically report only sum FA compositions of lipid species. In the present study, we examined both the glycerophospholipid (GPL) profiles and the comprising molecular lipid species in children with SAM before and 3 months after treatment with RUTF. It is known that the balance between saturated (SFA), monounsaturated (MUFA) and PUFA within phospholipids (PL) plays a role in sustaining the optimal biophysical properties of cellular membranes [13]. Thus, perturbation in this balance due to malnutrition as well as changes in response to RUTF treatment would in either case impact the crucial cellular processes (i.e., trafficking, function of membrane-anchored proteins, etc.) [13]. It has been demonstrated in other studies that fatty acids (FA) from the diet can redistribute within PL in a very selective manner, with phosphatidylcholine (PC) being the preferred sink for this redistribution [14]. Thus, investigating the fatty acid redistribution within PL in response to a dietary treatment “RUTF” would be advantageous to better visualize the involved metabolic pathways producing the fatty acyl constituents, the lipid classes and the pathways involved in lipid metabolism and turnover. We also studied the impact of different factors potentially associated with metabolic changes, including child age at enrollment, anthropometry, sex, breastfeeding, and hemoglobin levels.

## 2. Materials and Methods

This study was conducted as a secondary outcome exploration to a nutritional stepped wedge intervention study performed on children aged 6–36 months in Dadu, a rural district in Pakistan (Trial Registration Number: NCT00941434, www.clinicaltrials.gov). The people of this district are, in general, poor and experience high levels of mortality, morbidity, and disability. Briefly, a total of 888 children were recruited, who were divided into 16 clusters. The study started with 4 intervention clusters, and every quarter 4 clusters moved from control to intervention, thus in the last quarter all 16 clusters received the intervention. The sequence of rolling out clusters was randomized. Eligible children for participation in the study were categorized according to WHZ into 3 groups: (a) mild or no malnutrition WHZ ≥ − standard deviations (SD); (b) moderate (−3 < = WHZ < −2 SD); and (c) severe malnutrition (WHZ < −3). In this work, we focused on studying the metabolic impacts of treatment with RUTF on lipid metabolism only in children with SAM (WHZ < −3).

Severely malnourished children were treated with Plumpy’nut^®^ (Nutriset, Malaunay, France), a RUTF commonly used in treatment of SAM. Nutrition education regarding infant and young child feeding (IYCF) guidelines was provided to all families. At the time of enrollment, informed consent was obtained from parents or legal guardians. Each child was examined by a study physician. Very sick children were referred to the Public Civil hospital Dadu for inpatient treatment. Those children were kept under treatment till their WHZ normalized. At the time of recruitment, children were given a 15 days’ supply of Plumpy’nut^®^ according to the child’s current weight with a target dose of 200 kcal/kg&day [15]. Parents were given a 2 weeks’ supply in fortnightly visits with a body weight adapted dose and were instructed on usage/storage of these commodities. They were also advised to keep and return empty wrappers of the intervention to the study staff on their fortnightly visits for the purpose of assessing compliance with consumption of RUTF. Children were followed on a fortnightly basis and data were collected on anthropometry, morbidity/mortality, and compliance of commodities. The detailed relative fatty acid (FA) composition of Plumpy’nut^®^ was determined by gas chromatography (GC) as described by Jones et al. (2015) [16]. The results of the relative FA composition as well as the nutrient content per portion of 92 g [10] are presented in Table 1. During the follow up, children were deemed to be cured if their WHZ scores became ≥ −2 SD. For each participant, data on age, sex, demographics, height, weight, were recorded on the case report form and the fortnightly follow-up forms. Additionally, data on co-morbidities was defined as a child suffering from (1) At baseline: diarrhea and/or repeated episodes of cough/flu/sore throat and/or pneumonia and/or measles; (2) after treatment: current/14-day retrospective diarrhea and/or cough/flu/sore throat and/or fever and/or vomiting.

Dietary data including breastfeeding were recorded based on a 24 h dietary recall (24 h) for the food consumed by the children in the previous 24 h. The food group lists were defined based on a questionnaire from The World Health Organization (WHO) [17] and were aggregated into 7 groups: (1) cereals, roots, and tubers; (2) legumes and nuts; (3) dairy products (milk, yogurt, cheese); (4) flesh foods (meat, fish, poultry, and liver/organ meats); (5) eggs; (6) vitamin A rich fruits and Vegetables; (7) other fruits and vegetables. For breastfeeding, information was provided on whether the child was ever/currently breastfed, and if so with what frequency over the previous 24 h.

Metabolomic analyses were performed on plasma samples obtained from a subgroup of children with SAM with samples, anthropometric and laboratory data available, both at baseline and 3 months post-treatment.

### 2.1. Metabolomics Measurements

#### 2.1.1. Total Glycerophospholipid Fatty Acid Analysis

Total GPL fatty acids (GPL-FA) in plasma samples were analyzed using GC as described previously [18]. In brief, proteins of samples were precipitated with methanol containing internal standard (1,2-dipentadecanoyl-sn-glycero-3-phosphocholine). After centrifugation, sodium methoxide solution (25% in methanol) was added to the supernatant to form fatty acid methyl esters (FAME) from GPL. The reaction was stopped by adding methanolic HCl (3 mol/L), and FAME were extracted with hexane. Individual FAME were quantified by GC with flame ionization detection (GC-FID) (7890A; Agilent Scientific Instruments, Santa Clara, CA., USA). For the quality control (QC), six plasma QC samples were analyzed along with the samples. Concentrations at both time points were calculated in μmol/L, then the corresponding percentages were calculated relative to the total FA concentration.

#### 2.1.2. Analysis of Individual Phospholipid Species and Acyl Carnitines

Polar lipids including PL species of diacyl-phosphatidylcholines (PC.aa), acyl-alkyl-phosphatidylcholines (PC.ae), sphingomyelines (SM), acyl-lysophosphatidylcholines (lyso.PC.a), alkyl-lysophosphatidylcholines (lyso.PCe) as well as free carnitine (Carn) and acylcarnitines (Carn.a) were analyzed using flow-injection analysis tandem mass spectrometry (FIA–MS/MS) as described elsewhere [19]. In short, plasma samples (10 μL) were diluted with 500 μL methanol, containing internal standards for different lipid species and ammonium-acetate buffer. Samples were then centrifuged and 200 μL of the supernatant was transferred in 96-well plates prefilled with methanol (700 μL) for subsequent analysis. Analysis was performed using a liquid chromatographic system (Agilent 1200; Agilent Scientific Instruments, Santa Clara, CA., USA) coupled with a triple-quadrupole mass spectrometer (4000 QTRAP; Sciex, Framingham, MA, USA) with an electrospray ionization (ESI) source, operated in both positive and negative modes. Tandem mass spectrometry analysis (MS/MS) was performed in multiple reaction monitoring (MRM) mode. As mentioned earlier, six QC samples per batch were consistently measured with the samples. Concentrations were calculated in µmol/L; the analytical process was controlled and post-processed by Analyst 1.5.1 and R software (R Project for Statistical Computing, http://www.r-project.org/). A formula CX:Y was assigned for polar lipids where X indicates the number of carbon atoms in fatty acids, Y the number of double bonds, and OH the presence of a hydroxyl group. Letters ‘a’ and ‘e’ indicate that the acyl chain is bound via an ester or ether bond to the backbone, respectively.

### 2.2. Quality Control (QC) and Preprocessing

QC for metabolomics measurements was performed as previously described [20]. For GPL-FA, we calculated the sums of SFA (∑SFA), MUFA (∑MUFA), PUFA (∑PUFA) and the ratios: Mead acid (C20:3n-9) to arachidonic acid (AA, C20:4n-6) (marker of EFA deficiency) [21,22]; AA to linoleic acid (LA) (C20:4n-6/C18:2n-6) (a marker of PUFA to LC-PUFA conversion by Δ-6 desaturase, Δ-5 desaturase, and elongase activities) [23,24]; eicosapentaenoic acid (EPA, C20:5n-3) to AA; n-6 docasapentaenoic to docosahexaenoic acid (n-6 DPA/DHA, C22:5n-6/C22:6n-3): and n-6 to n-3 PUFA (markers of n-3 PUFA status) [25,26]. For the PL species, we calculated ∑PC.aa, ∑PC.ae, ∑lyso.PC.a, ∑lyso.PCe, ∑PC, and ∑SM.

### 2.3. Statistical Analysis

Statistical analysis was performed independently on the two obtained data sets (GPL-FA and polar lipids [PL and carnitines]), each composed of paired sample data for severely malnourished children at two time points (baseline and after 3 months of treatment with RUTF). Normally distributed variables are described by mean ± SD and non-normally distributed variables by median and interquartile range (IQR). Normality was tested with Shapiro–Wilk (S-W) test. For exploratory data analysis, unadjusted comparisons using multiple univariate tests were performed within the Metaboanalyst 3.0 software which include fold change (FC) analysis, two-tail paired Student *t*-test or Wilcoxon rank-sum test for normal or non-normal distribution, respectively, and a combination of FC and t-test/Wilcoxon rank-sum test to produce a volcano plot [27]. A FC threshold of 1.5 and a *p*-value of 0.05 (adjusted for multiple testing using false discovery rate [FDR]) [28] were selected to identify the significantly different features (metabolites) between the two groups (pre- and post-treatment) using the FC analysis and *t*-tests, respectively. Then, important features recognized by both tests were depicted in volcano plots with |log10 (*p*)|-values and log2 (FC) on y- and *x*-axis, respectively. On the other hand, for class discrimination and identification of metabolites that most drive group separation, several multivariate tests were also attempted including: principal component analysis (PCA), partial least squares- discriminant analysis (PLS-DA) and orthogonal partial least squares–discriminant analysis (orthogonal PLS-DA), high-dimensional feature selection–significance analysis of microarrays (and metabolites), empirical Bayesian analysis of microarrays (and metabolites) (EBAM) as well as random forests (RF).

Another aspect of the analysis involved identifying predictors of the change in the metabolite levels whether the GPL-FA (ΔFA; FA final – FA baseline) or polar lipids data sets (ΔPL; PL_final_–PL_baseline_). For this purpose, linear regression models were generated using log2 of the change in metabolite concentration as outcome and each of potential confounders (age, sex, WHZ z-scores and Hb concentration, and breastfeeding at baseline) as well as the metabolite concentration at baseline as independent variable and adjusting for the remaining confounders. *p*-values (FDR adjusted) <0.05 were considered statistically significant.

## 3. Results

### 3.1. Characteristics of the Study Population

We included 41 children with SAM (56% males) in the study, with an age of 21.8 ± 6.5 months (mean ± SD) and weight for height z-scores (WHZ) of -3.56 ± 0.56 at baseline (characteristics of the study population are shown in Table 2). Thirty four percent of the children were breastfed. Treatment with RUTF achieved a significant increase in length, weight, WHZ z-scores and Hb values after 3 months. Out of the 41 children with available GPL-FA data, 23 (56%) children were ill at baseline of which 14 (34%) were with diarrhea, 19 (46%) were with cough/flu/sore throat, 1 (2.4%) were with pneumonia and none were with measles. No significant difference was detected in co-morbidity between baseline and after 3 months of RUTF treatment.

Regarding the dietary intakes, information on the study participants (%) by food group, both at baseline and after treatment with RUTF for 3 months are provided in Table 3. Based on the data, at baseline, all the children consumed a diet based on grains, roots and tubers with only 4.9%, 12.2% and 14.6% of them consuming flesh foods, vitamin A rich fruits and vegetables, and other fruits and vegetables, respectively. The results showed no significant differences in the children’s consumption for all the food groups between baseline and after 3 months of RUTF treatment.

### 3.2. Changes in Lipid Metabolites after Treatment with Ready-to-Use Therapeutic Foods (RUTF)

At baseline, the malnourished children showed relatively low plasma values (median [IQR] % *w*/*w*) of LA (16.84[5.16]), alpha linolenic acid (ALA) (0.15[0.11]), and even lower values of long-chain PUFA (LC-PUFA) such as AA (4.19[2.07]) and DHA (0.68[0.41]. These values are much lower than EFA values in healthy and well-nourished children investigated with the same analytical method [18] who showed values for LA of 22.27[4.22], for AA of 9.64[2.73], and for DHA of 5.82[2.05] (Appendix A), indicating a depletion of EFA and proportionally even more LC-PUFA. EFA depletion was reflected in the higher Mead acid/AA in malnourished children at baseline compared to healthy children, which was almost four times higher than its corresponding levels in healthy children. An apparent decline in the conversion of EFA to the long-chain PUFA metabolites was reflected in the lower values of AA/LA which were approximately half of the corresponding values in well-nourished children investigated with the same method [18]. FC analysis showed an increase after treatment above the chosen threshold (FC = 1.5) in ALA (C18.3n-3), C22.6n-3, and a decrease in Mead acid/AA, which indicate an EFA replenishment, especially n-3 PUFA, however the values did not completely normalize as compared to the ranges in the healthy children, investigated with the same method [18]. After treatment, several FA and FA classes were significantly different from baseline (paired *t*-test/Wilcoxon rank sum test), including SFA (C16.0, C17.0, C18.0), PUFA (n-6, C18.2n-6, C20.4n-6, C22.4n-6, C22.5n-6, and n-3, C18.3n-3, C22.5n-3, and C22.6n-3), sums (∑SFA, ∑n-3 PUFA, and ∑ PUFA), and ratios (n-6 PUFA/n-3 PUFA, Mead/AA, C22.5n-6/DHA and AA/LA) (*p*_ FDR_ <  0.05) (Table 4A). As shown in the volcano plot (Figure 1A), only C18.3n-3 and C22.6n-3 were significantly increased while Mead: AA was significantly decreased.

With regard to the changes in plasma individual polar lipid species (PL and carnitines) after treatment, as evident in the FC analysis results, lyso.PC was the PL species most impacted by the treatment. ∑lyso.PC including both ∑lyso.PC.a and ∑lyso.PC.e significantly decreased beyond the selected threshold reflecting the decrease in numerous individual lyso.PC species (lyso.PC.a.C16.0, lyso.PC.a.C18.0, lyso.PC.a.C16.1, lyso.PC.a.C18.1, lyso.PC.a.C18.3, lyso.PC.a.C14.0, lyso.PC.e.C16.0, lyso.PC.e.C18.0). Other PL species decreased beyond the selected FC threshold, including SM.a.C30.1, PC.ae.C36.1, PC.ae.C36.0, PC.ae.C34.0, PC.ae.C30.0, PC.ae.C40.2 and PC.aa.C32.1 as well as one Carn.a, namely Carn.a.C8.1. The same picture was confirmed in the paired *t*-test/Wilcoxon rank-sum test of values before and after treatment, where ∑lyso.PC, ∑lyso.PC.a and ∑lyso.PC.e were the significantly different PL species, in addition to few PC and SM (Table 4B). The volcano plot (Figure 1B), shows that several individual lyso.PC reflected in ∑lyso.PC.e, ∑lyso.PC.a, and ∑lyso.PC and a few PC were significantly reduced while only Carn.a.C8.1 was significantly increased after treatment.

When applying the feature identification using significance analysis of microarrays or EBAM, whether on GPL-FA or PL data, all metabolites recognized as significant by the *t*-test were also found significant by these two tests using a default delta of 1.2 and 0.9, respectively (Appendix A). All the other attempted multivariate models on both datasets were not able to fully discriminate samples before and after treatment with RUTF, except for RF using (number of trees: 500; number of predictors:7) which showed an OOB (out-of-bag) RF classification error of 0.183 and 0.305, respectively. In the total GPL-FA, C16.0, PUFA, Σn-3 PUFA, and C22.5n-3 were ranked as the most important variables discriminating samples before and after treatment (Figure 2A) while lyso.PC.e.C18.0, PC.ae.C34.0, lyso.PC.a.C18.1, lyso.PC.a.C16.0, ∑lyso.PC.a followed by PC.aa.C38.6, PC.ae.C30.0 and PC.ae.C36.1 were selected from the PL data (Figure 2B). Almost the same discriminating species were recognized by both the univariate and multivariate analysis, for further confirmation of the findings.

Some PC and SM were elevated, while others were decreased post-treatment especially those comprising SFA and MUFA (PC.ae.C36.1 (PC(o-16.1(9Z)/20.0)); PC.ae.C36.0 (PC(o-16.0/20.0)); PC.ae.C34.0 (PC(o-16.0/18.0)); PC.ae.C30.0 (PC(o-14.0/16.0)); PC.ae.C40.2 (PC(o-18.2(9Z,12Z)/22.0)); PC.aa.C32.1; (PC(o-16.0/16.1(9Z)); SM.a.C30.1; SM.a.C31.1; SM.a.C32.1; SM.a.C33.1; SM.a.C34.1, SM.a.C35.0; SM.a.C35.1; SM.a.C36.1, SM.a.C38.1; SM.a.C39.1; SM.a.C41.1; SM.a.C42.1). ΣPCs and ΣSM showed a non-significant trend to a decrease (Appendix A), potentially due to a higher contribution of species comprising SFA and MUFA to ΣPC and ΣSM.

### 3.3. Correlates of Change in Metabolite Concentrations after Treatment

To facilitate the interpretation of associations between potential confounders and changes in the metabolite concentrations, we determined the direction of change of each of the metabolites and metabolite groups in the two investigated datasets. Appendix A shows the median and IQR of the FA and FA groups before and after treatment. Box plots showing differences in PL concentrations before and after treatment are presented in Appendix A. The results from linear models for all investigated metabolites are shown in Appendix A. We focus in our results and discussion sections on metabolites with significant differences after FDR correction (*p*-value_FDR_ < 0.05). Both child age and breastfeeding status at baseline showed a significant impact on the ΔFA percentages, as evident in a large number of FA species significantly associated with these 2 factors (Table 5 and Table 6). Both older age and breastfeeding at baseline were positively associated with ΔSFA and especially C18.0, and inversely with ΔMUFA and especially C18.1n-9 (Table 5 and Table 6). In addition, child age at baseline was inversely associated with the change in C15.1, C16.1n-7, C22.1t, C14.0 while positively associated with the increase in C20:3n.6 (Table 5). While breastfeeding was negatively associated with the increase in ALA (Table 6), it tended to be positively associated with Σn-6 PUFA and ΣPUFA (not significant after FDR correction) (Appendix A). Child age and breastfeeding were also associated with numerous PL species as well as ΣPC, PC.aa and PC.ae (Table 5 and Table 6).

Baseline hemoglobin showed a much lesser association with ΔFA where it was inversely associated with C18.2.tt and C17.0 (Table 7A), while it was not significantly associated with other FA (Appendix A). In contrast, Hb was negatively associated with the concentration changes of several PL species including ΣPC.ae, ΣPC.aa, and ΣPC (Table 6B). WHZ z-scores and sex were not associated with changes in GPL and PL species, accordingly they were removed from the models.

## 4. Discussion

### 4.1. Changes in Lipid Metabolites after Treatment with RUTF

Our study confirms that children with SAM are depleted in EFA and to an even greater extent in LC-PUFA metabolites, which confirms previous reported data [29]. Compared to healthy children, the untreated malnourished children showed lower plasma glycerophospholipid contents of LA, ALA, LC-PUFA, as well as reduced markers of EFA conversion to LC-PUFA (AA/LA), but higher values of Mead acid/AA, a marker of EFA deficiency [21,22]. EFA depletion is a hallmark of malnutrition which may result from a combination of factors, including low dietary fat and EFA supply [30], reduced gastrointestinal digestion and absorption of dietary lipids, enhanced beta-oxidation of FA for ATP formation, and enhanced lipid peroxidation [11,29]. The dietary pattern could be one of the reasons underlying the low EFA supply in those children at baseline. This hypothesis is supported by the dietary data of the children in this study (Table 3) demonstrating that all the children consumed a diet based on grains with only a low percentage of children consuming foods, vitamin A rich fruits and vegetables, or other vegetables or fruits. This diet characterized by a lack of food diversity does not likely or perhaps barely meet the recommendations for n-6 PUFA, but not n-3 PUFA or n-6/n-3 ratio. The presented data matched with the consensus that diets in most of the low-income countries are based mainly on stable foods (cereals, legumes and roots) which generally have a low content of PUFA, especially n-3 PUFA [31]. For Pakistani households, cereals are the main source of calories (60%) followed by 12% from oils, and 10% from sugars [32]. Apart from this, the energy density of a diet based on cereals is low which means that the food is too bulky, and the child will not be able to eat adequate amounts. Infants and young children have a limited gastric capacity and an energy requirement (/unit body weight) about three times as high as adults. Therefore, even non-malnourished children given a low-energy density diet may not be able to eat adequate amounts because of the bulkiness of the diet [33].

Infants/young children in low-income countries could be also more prone to low birthweights and thus poor fetal stores, which could make them vulnerable and dependent on a postnatal dietary supply of LCPUFA which if not sufficiently provided in diet would lead to EFA depletion [33]. Another hypothesis involves exposure of children in poor or low-income countries to environmental stress (e.g., infections) which increases their requirement for PUFA, as previously reported [26,33,34,35,36,37]. To test this hypothesis, we investigated the associations between baseline co-morbidity and GPL-FA which showed no significant associations with the GPL-FA, at baseline after adjustment for age and sex, except for C20:3n-6 and C18:3n-6 whose values tended to be lower in ill children at baseline (not significant after FDR correction) (Appendix A). Therefore, the present findings do not sufficiently support the impact of the infection state on the GPL-FA profiles including PUFA.

The resulting depletion of EFA appears to be aggravated by an additional reduced conversion of EFA to LC-PUFA, which may result from limited intracellular availability of energy and acetate units required for the EFA conversion to their corresponding LC intermediates as well as cofactors of desaturating enzymes such as zinc. Low LC-PUFA may induce adverse effects on membrane integrity, growth, immune and neural system functions [38].

Treatment with RUTF for 3 months significantly improved EFA and LC-PUFA status and led to a significant increase of the AA/LA ratio. The levels achieved, however, did not reach the normal ranges in healthy children (Appendix A), even though the malnourished children were generally deemed to be cured (WHZ z-scores ≥ −2 SD) (Table 2). The detected increase in the EFA as well as LC-PUFA post-treatment towards normal levels might be expected, since the RUTF formula is a source of EFA and also provides the energy and substrates needed for the biosynthesis of LC-PUFA. Plumpy’Nut is rich in LA (12.62%) and provides ALA (1.11%). Another hypothesis would be the supply of the deficient LC-PUFA through the formula itself. According to the reported composition of Plumpy’Nut, it contains a considerable portion of AA (0.98%), thus the increase in AA values after treatment could have been facilitated by its supply with the formula (Table 1). However, previous reports on RUTF composition showed no appreciable amounts of other preformed LC-PUFA [39,40,41,42].

The increasing levels of PUFA and particularly of AA after treatment were associated with a decrease in the Mead acid/AA ratio, which reflects the repletion of intracellular EFA levels after treatment. Mead acid/AA ratio, a recognized marker of EFA deficiency, showed higher values in malnourished children at baseline relative to heathy children, as mentioned in the results section. In healthy subjects, LA competitively inhibits δ-6-desaturation of oleic acid into n-9 Mead acid, while under the condition of low or deficient LA, the δ-6-desaturase enzymes uses C18.1n-9 as a substrate producing the Mead acid [21,22].

The increase of the ARA/LA-ratio suggests that RUTF treatment contributed to a correction of the initially impaired EFA conversion to LC-PUFA by consecutive desaturation and chain-elongation. Impaired EFA conversion to LC-PUFA was repeatedly reported in children with primary and secondary malnutrition [24,43,44,45,46,47]. The increase in DHA post-intervention was somewhat higher than expected relative to ALA, also considering the modest ALA supply with the RUTF. Therefore, we consider this increase not to be solely due to the increased EFA conversion to LC-PUFA, but there might be additional effects of reduced oxidative catabolism of DHA. Exchange and interactions between FA pools in different blood compartments e.g., red blood cells (RBCs) and plasma, might also have occurred. DHA was previously reported to decline more slowly than other n-3 and n-6 PUFA, maintaining higher basal values in plasma during a wash out phase after 18 weeks treatment with an n-3 PUFA supplement, presumably due to redistribution from body pools [48]. It is also worth noting that the amount of LA provided was 12 times greater than that of ALA, but post-intervention the relative increase of DHA was even higher than that of AA. This could be a result of the competition between ALA and LA conversion to the corresponding LC-PUFA, where ALA is the preferred substrate for the enzymes involved in the conversion process.

Although RUTF is rich in SFA, comprising almost 35% of the total FA content, ∑SFA decreased after treatment, in agreement with previous reports in different types of malnutrition [29,49,50,51]. Malnutrition promotes lipolysis of adipose tissue triglycerides to provide an energy source, which releases SFA and particularly palmitic acid (C16.0). This hypothesis is supported by the fact that PA is the most common SFA whether in the adipose triacylglycerols (TAG) or in membrane PL, representing 20–30% of total fatty acids (FA) [52]. In fact, increased levels of non-essential SFA and MUFA in all lipid classes have previously been reported in untreated malnutrition [29]. During recovery, due to the increased availability of lipids and proteins in RUTF, lipolysis decreases promoting fat deposition, and thus a decrease in circulating SFA. In general, the elevation in the percentage of PUFA at the expense of SFA in the erythrocyte membranes and whole blood in response to RUTF treatment, was previously reported [16,25].

The composition of PL species mirrors to some extent the changes in total GPL-FA. This was most evident in the decrease of PL comprising SFA and MUFA and particularly lyso.PC. Lyso.PC shows the greatest change of all PL in response to the intervention, with a decrease of saturated (lyso.PC.a.C14.0, lyso.PC.a.C16.0, lyso.PC.a.C18.0, lyso.PC.e.C16.0, lyso.PC.e.C18.0) and monounsaturated lyso.PC (lyso.PC.a.C16.1, lyso.PC.a.C18.1) (Figure 1B). These lyso.PC, specifically lyso.PC containing 16:0, 18:0, 18:1, were also previously shown to be among the most abundant circulating serum human metabolites [53] in agreement with our results in plasma, where ∑lyso.PC represent ≈36% of total PL, while lyso.PC.a.C16.0 and lyso.PC.a.C18.0 combined comprise ≈33% of total PL. Previous reports suggested that lyso.PC are secreted by the liver as a product of hydrolysis of hepatic PC [54]. A major source of plasma lyso.PC is the lecithin cholesterol acyl transferase (LCAT) reaction which hydrolyses PC at the sn-2 position, mostly occupied by unsaturated FA thus generating saturated lyso.PC [54]. On the other hand, other studies reported after treatment of malnutrition an increase in several lyso.PC, principally the major ones, saturated lyso.PC.a.C18:0 and lyso.PC.a.C16:0 [11].

ΣPC and ΣSM showed a non-significant trend to a decrease, predominantly due to lower levels of PC and SM comprising SFA and MUFA (Appendix A). Although some PL comprising EFA and LC-PUFA increased with the RUTF treatment, we did not notice a specific significant increase in the PC species comprising LA, ALA, AA provided in the formula. In fact, ∑PC, ∑lyso.PC, ∑SM tended to be lower, indicating that the decrease in species comprising SFA and MUFA had a more marked effect on total PC. The differences in the concentrations of individual PL species are depicted in the boxplots (Appendix A). Our PL analysis did not include some potentially relevant PL species such as phosphatidylserines (PS), phosphatidylethanolamines (PE), thus it is conceivable that the detected PUFA changes are influenced by alterations in these species. One aspect that complicates the interpretation is that FA are differentially incorporated into different blood compartments and lipid classes [24,55]. Only few previous studies in relation to malnutrition presented a complete list of FA, while many studies studied only the major FA [24,56] and others focused only on the n-6 EFA [49,57].

Medium and long-chain Carn.a tended to increase after treatment (not significant), with few exceptions (Carn.a.C14.0 and Carn.a.C18.0) (Appendix A). Carn.a.C8.1 was the only Carn.a that was significantly increased (Figure 1B). These results were in agreement with previous reports showing an increase in the level of Carn.a comprising LA, AA, medium chain FA, and Carn.a.C3.0 after treatment with RUTF [12]. This might be a consequence of decreased Carn synthesis as a result of impaired protein breakdown and shortage of the amino acid precursors of Carn, lysine and methionine [58]. This hypothesis matches our findings of a non-significant trend to higher levels of Carn after treatment (Appendix A). Some reports proposed impaired peroxisomal oxidation of lipids as one of the metabolic disruptions encountered in malnutrition [59,60], but this might also be a metabolic adaptation to the decreased FA availability [49]. As shown in the previous findings, remarkable changes were detected in the concentrations of GPL-FA species and their comprising molecular species. These changes may be attributed to the consumption of the RUTF formula however it might also be a result of other factors (change in the clinical status, consumption of other foods, changes in the relative quantities of breast milk consumed relative to the study formula) or a combination of two or more of these factors. By examination of the parameters with available data, we found no significant differences in the comorbidities, consumption of the food groups, frequency of breastfeeding after treatment relative to baseline data, as previously shown in the results section. In addition, the results showed lack of significant associations between the changes in the GPL-FA profiles and the disease status at baseline. On the other hand, we have no available data on the relative consumption of breastmilk relative to the study formula. Therefore, we interpreted the findings presumably a result of the consumption of the RUTF formula, however a possible contribution of other factors cannot be excluded.

### 4.2. Correlates of Change in Metabolite Concentrations after Treatment

We found age and breastfeeding impact GPL-FA levels. Older children had higher baseline SFA and MUFA with lower PUFA percentages, principally n-3 PUFA, and they showed a more pronounced decrease in SFA and a lesser decrease in MUFA after treatment. This might reflect the consequences of longer duration of exposure to a poor dietary PUFA supply and hence a greater degree of depletion of body stores at an older age. Child age was also positively associated with changes in numerous PL classes, principally with the decrease in ∑PC, ∑lyso.PC, and ∑SM. Since older children showed higher SFA levels at baseline, which represent >49% of total GPL-FA versus 25% for MUFA and 27% for PUFA, it is not surprising that the greater decrease in SFA is reflected in a higher decrease in the sums of these PL species (PC, Lyso.PC) (Table 5) as well as SM where also SFA contributes largely to its composition.

Breastfeeding was also associated with the same FA classes in the same directions, and also with a trend to higher Σn-6 PUFA and ΣPUFA (not significant after FDR correction). Breastmilk provides LC-PUFA, including DHA and AA, to generally meet the needs of the developing baby for development of visual acuity and neural functions [61]. Despite the lack of available data on the PUFA composition of the breastmilk consumed by the children in this study, we speculate that the LC-PUFA supply with breastfeeding might provide an explanation for the non-significant trend to a greater increase in LC-PUFA, such as AA and C20.3n.6, and DHA in breastfed infants after intervention, as compared to non-breastfed infants. This speculation is in line with the established consensus that breastmilk is the ‘gold standard’ for most nutrients [62,63]. Provided the optimal maternal living and nutritional conditions, breastmilk alone provides not only sufficient energy supply (50 ± 60% of total energy) from fat, but also all LC-PUFA and their parent n-6 and n-3 EFA necessary to meet the infant’s requirements for normal development, at least up to 4 ± 6 months of age [64,65]. Previous studies have reported lower levels of LC-PUFA in formula fed babies in comparison to breastfed infants [66,67]. In other studies, close associations have been observed between the breastmilk fatty acid composition infant plasma, RBCs [68], and tissue FA levels [69,70,71]. In another study, a positive correlation was found between n-6/n-3 LC-PUFA ratio in breastmilk at 4 months of lactation and infant RBCs at 4 and 12 months of age [72].

On the other hand, the breastmilk composition was proven to be influenced by maternal diet, especially under conditions where uncertainty regarding the appropriate maternal diet are involved [62]. Perhaps, this would explain why the trends towards higher PUFA observed in breastfed infants in this study were much less pronounced for ∑n-3 PUFA. In North Pakistan, low breastmilk n-3 LC-PUFA levels might be expected because of the predominant use of corn oil and ghee which are low in ALA and do not provide DHA, a low intake of green leafy vegetables that serve as a source of ALA, and lack of consumption of n-3 LC-PUFA-rich fish [62,73].

In contrast to our findings, another study reported lower levels of LA in association with breastfeeding [25]. Human milk lipid FA comprise approximately 45–50% MUFA [61]. This large proportion of MUFA may explain the lower decrease of MUFA in the breastfed infants after treatment with RUTF, as compared to non-breastfed infants. Regarding the impact of breastfeeding on PL species, an overall picture remarkably similar to the impact of age was obtained. The positive association of breastfeeding with the decrease in ∑PC, especially ΣPC.aa might reflect the detected higher decrease in SFA, which represent the major FAs comprising the membrane PL [52]

While baseline Hb was significantly associated with a few FA species, a higher gain in Hb (ΔHb) was associated with a higher increase in PUFA, especially n-6 PUFA. Perhaps, this is the reason behind the higher decreases seen in numerous PC species in anemic subjects. This increase in n-6 PUFA is associated with a decrease in SFA, a major contributor to PC resulting in a higher decrease in ΣPC.

No association was found between WHZ z-scores with the change in total GPL-FA composition or PL species after treatment. The same applies to PL data, where only a trend towards negative association was found between the gain in WHZ z-scores and the decrease in PC.aa.C36.1, PC.ae.C40.1 and similarly a positive trend with the decrease Lyso.PC.a.20.4, Carn.a.C9.0. However, none of these associations were significant after FDR correction. Underweight and stunting was repeatedly linked to depletion of EFA and their corresponding LC metabolites (LC-PUFA) in previous studies. Also, in previous studies involving children recovering from malnutrition, weight gain was found to be positively associated with n-6 PUFA levels in whole blood, especially LA [21]. According to our results, sex showed no impact on the change in GPL-FA and individual PL species in response to treatment with RUTF. These findings were in accordance with some previous reports which showed that no significant differences were found between sexes in individual FA [74]. On the other hand, they were in contrast with others that referred to sex-specific differences in EFA metabolism between males and females [25,75,76].

## 5. Strengths and Limitations

Only few studies have addressed changes in the lipid and metabolomics profiles in response to nutritional intervention in malnourished children [11,12], with other studies quantifying changes in FA but not the PL species [25,40,49]. We investigated both GPL-FA composition and individual PL species with a state of the art quantitative, targeted metabolomics platform, quantifying changes in FA composition and their incorporation and distribution in different PL fractions (PC, Lyso.PC, and SM) in response to RUTF treatment. A limitation of the study is the limited number of children that could be studied and the lack of a healthy control group from the local population, which was not included for ethical reasons, because it is difficult to justify burdening healthy children with repeated blood sampling. Rather, we had to revert to comparing data with those from healthy children in Europe. Also, although GPL-FA covered the FA species comprising all PL species, the employed PL analyses were limited to PC and did not include some potentially relevant species such as PS and PE. Another limitation is the lack of data on the clinical auxologic and cognitive effects relevant to the changes in lipid species after RUTF treatment, which makes the interpretation of the physiological impacts of changes in lipid species compositions rather difficult. Data on quantitative dietary food intakes and relative intakes of the breastmilk relative to the study formula were also lacking which is regarded as a drawback of the study.

## 6. Conclusions

Malnourished children have decreased blood levels of EFA, LA and ALA, and to an even greater extent of their corresponding n-6 and n-3 LC-PUFA metabolites, with an apparent impairment in the endogenous conversion of EFA to LC-PUFA. All of these improve with RUTF treatment. Treatment induced a trend to a decrease in the PL (∑PC and ∑SM) and a significant decrease in ∑lyso.PC, which appears to reflect the decrease in SFA as result of metabolic stabilization. Breastfeeding provides protective effects with respect to supporting higher n-6 PUFA and LC-PUFA values than found in non-breastfed infants. Older children showed a greater decrease in SFA and a lesser decrease in MUFA with treatment reflected in a higher decrease in ∑PC, ∑lyso.PC as well as ∑SM, which may be due to a greater depletion of PUFA body stores in older children exposed for longer time periods to an inadequate dietary supply. In spite of the fact that there was a marked improvement detected in the WHZ z-scores and Hb after RUTF treatment, no evidence was found relating this outcome to the detected changes in the GPL-FA levels and their comprising molecular species. The methodology used appears suitable for evaluating metabolic effects of dietary treatment in SAM and may be used for further evaluation of improved intervention strategies.

## Figures and Tables

**Figure 1 nutrients-12-02163-f001:**
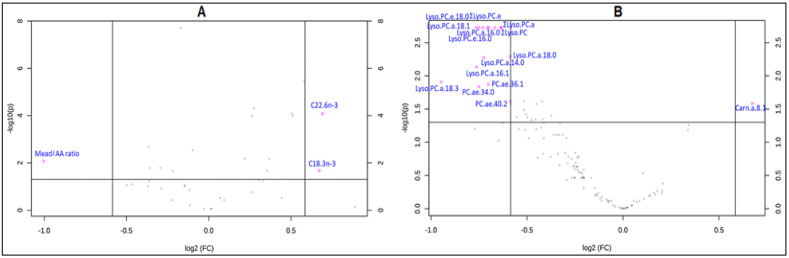
Volcano plot depicting metabolites with significant changes during intervention: (**A**) total glycerophospholipid fatty acids (**B**) phospholipid species with fold change (FC) threshold of 1.5 and t-tests threshold of 0.05. The log transformed fold changes and the *p*-values are represented on x- and *y*-axis, respectively. The red circles represent features above the selected thresholds. Metabolites lying further away from the (0,0) are more significant.

**Figure 2 nutrients-12-02163-f002:**
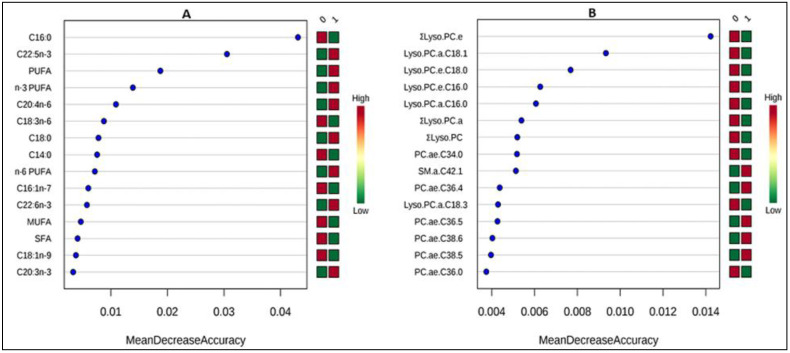
Significant metabolites: (**A**) total glycerophospholipid fatty acids (**B**) phospholipid species identified by random forest. The metabolites are ranked by the mean decrease in classification accuracy when they are permuted. Red and green squares indicate high and low levels of the metabolites, respectively. “0” and “1” refer to the two time points investigated in the study (t0 or baseline and t1 (after treatment with RUTF).

**Table 1 nutrients-12-02163-t001:** Fatty acid (FA) composition (% wt/wt) and nutrient content/portion (92 g) of Plumpy’nut^®^, the ready-to-use therapeutic food (RUTF) used in treatment of severe acute malnutrition (SAM) in this study. Data from [16] and [10].

Fatty Acid	%
C14:0	0.75
C15:0	0.05
C16:0	28.82
C16:1n-7	0.16
C18:0	3.61
C18:1n-9	48.57
C18:1n-9t	1.03
C18:2n-6	12.62
C18:3n-6	0.06
C18:3n-3	1.11
C20:0	0.55
C20:1n-9	1.03
C20:2n-6	0.03
C20:3n-6	0.00
C20:4n-6	0.98
C20:4n-3	0.09
C20:5n-3	0.04
C24:0	0.45
C24:1n-9	0.06
C22:5n-3	0.00
C22:6n-3	0.00
Σn-6 PUFA	13.68
Σn-3 PUFA	1.24
n-6: n-3 PUFA	11.05
n-6 (% energy)	7.25
n-3 (% energy)	0.66
**Nutritional Value:**	**g/92 g**
Carbohydrates	45
Proteins	12.8
Fat	30.3
Energy (kJ/kcal)	2100/500

Data are expressed as weight percentage of total fatty acids. PUFA, polyunsaturated fatty acids.

**Table 2 nutrients-12-02163-t002:** Characteristics of the population included in the study at baseline and after treatment with RUTF.

Characteristics	At Baseline	After Treatment	*p*-Value
Sex	Males (22; 54%), Females (19; 46%)		
Breastfeeding status (y/n)	14 (34%)	14 (34%)	ns
Hemoglobin (Hb; (g/dL))	8.35 ± 2.13 [6]	9.71 ± 1.52 [7]	**
Length (cm)	77.14 ± 8.09	78.94 ± 7.90 [4]	***
Weight (kg)	7.36 ± 1.45	8.78 ± 1.67 [4]	***
Weight for height z-scores (WHZ)	−3.56 ± 0.56	−1.91 ± 0.76 [7]	***
Co-morbidity (y/n)	23 (56%)	18 (53%) [7]	ns
Diarrhea (y/n)	14 (34%)	14 (41%) [7]	ns
Repeated episodes of cough/flu/sore throat (y/n)	19 (46%)	11(32%) [7]	ns
Pneumonia (y/n)	1 (2.4%)	na	
Measles (y/n)	0(0%)	na	
Fever	na	12 (35%) [7]	
Vomiting	na	3 (8.8%) [7]	

Tabulated values are expressed in ‘mean ± standard deviation (SD)’. Numbers in square brackets indicate the numbers of missing observations. *p*-values are expressed as *** *p* < 0.001, ** *p* < 0.01, ns non-significant. NA not available. Paired differences between the child’s characteristics at baseline and after treatment with RUTF were evaluated via McNemar’s Chi-squared and Chi-squared tests for categorical and continuous variables, respectively.

**Table 3 nutrients-12-02163-t003:** Study participants (%) by food group at baseline and after treatment with RUTF.

Food Groups	Dietary Components	At Baseline (%)	After Treatment (%)	*p*-Value
1	Grains, roots and tubers	100	100 (7)	ns
2	Legumes and Nuts	12.2	20.6 (7)	ns
3	Dairy Products (milk, yogurt, cheese)	51.2	55.9 (7)	ns
4	Flesh foods (meat, fish, poultry, and liver/organ meats)	4.9	11.8 (7)	ns
5	Eggs	2.4	5.9 (7)	ns
6	Vitamin A rich fruits and Vegetables	12.2	14.7 (7)	ns
7	Other fruits and Vegetables	14.6	11.8 (7)	ns
8	Breastmilk consumption	5.4 ± 1.6	5.2 ± 1.1 (7)	ns

Data are expressed as percentage of total study participants by group according to 24 h recall rate. Breastmilk consumption is expressed as frequency of breastfeeds according to 24 h recall and is described by mean ± SD. Paired differences between the child’s characteristics at baseline and after treatment with RUTF were evaluated via McNemar’s Chi-squared and Chi-squared tests for categorical and continuous variables, respectively. *p*-values expressed as ns non-significant. Numbers between brackets indicate the numbers of missing observations.

**Table 4 nutrients-12-02163-t004:** Metabolites (**A**) Total glycerophospholipid Fatty acids (**B**) polar lipid species identified as significant features between lipid profiles of severe malnourished children (n = 41), pre- and post- treatment with RUTF.

Metabolites	Baseline	After Treatment with RUTF	*p*-Value (after False Discovery Rate (FDR) Correction))
**A. Fatty acids**			
**Saturated fatty acids:**			
C16:0	30.97 ± 1.96	27.49 ± 2.46	<0.001 ***
C17:0	0.56 ± 0.15	0.48 ± 0.13	0.01 *
C18:0	15.93[2.34]	17.73 [3.83]	0.02 *
ΣSFA	48.91 ± 3.81	45.74 ± 4.34	0.001 **
**Monounsaturated fatty acids:**			
C16:1n-7	1.22[0.83]	0.90[0.39]	0.005 **
**Polyunsaturated fatty acids:**			
**n-6:**			
C18:2n-6	16.40 ± 3.70	19.09 ± 3.85	0.002 **
C20:4n-6	4.19[2.07]	5.97[2.12]	<0.001 ***
C22:4n-6	0.38 ± 0.14	0.49 ± 0.17	0.002 **
C22:5n-6	0.33 ± 0.16	0.42 ± 0.15	0.009 **
Σn-6 PUFA	24.35 ± 4.61	29.21 ± 5.04	<0.001 ***
**n-3:**			
C18:3n-3	0.15[0.11]	0.19[0.16]	0.01 *
C22:5n-3	0.42 ± 0.17	0.59 ± 0.18	<0.001 ***
C22:6n-3	0.71 ± 0.37	1.15 ± 0.47	<0.001 ***
Σn3 PUFA	1.60 ± 0.58	2.38 ± 0.67	<0.001 ***
ΣPUFA	26.42 ± 4.82	31.95 ± 5.28	<0.001 ***
**Ratios:**			
n-6/n-3 -PUFA	16.79 ± 5.65	13.03 ± 3.36	<0.001 ***
Mead acid/AA	0.07[0.09]	0.03[0.05]	0.002 **
C22:5n-6/DHA	0.49[0.27]	0.40[0.15]	0.006 **
AA/LA	0.28 ± 0.10	0.34 ± 0.12	0.020 **
**B. Polar lipid species**			
**Lysophosphatidylcholines:**			
Lyso.PC.e.C16.0	1.04 ± 0.40	0.64 ± 0.45	<0.001 ***
Lyso.PC.e.C18.0	3.38 ± 1.58	1.99 ± 1.45	<0.001 ***
Lyso.PC.e.C18.1	0.35 ± 0.14	0.25 ± 0.18	0.003 **
ΣLyso.PC.e	4.77 ± 1.97	2.87 ± 2.02	<0.001 ***
Lyso.PC.a.C18.3	0.09[0.06]	0.04[0.06]	0.005 **
Lyso.PC.a.C14.0	0.32 ± 0.18	0.19 ± 0.13	<0.001 ***
Lyso.PC.a.C16.0	97.73 ± 35.09	61.55 ± 45.53	<0.001 ***
Lyso.PC.a.C16.1	0.44 ± 0.24	0.26 ± 0.23	<0.001 ***
Lyso.PC.a.C18.0	75.25 ± 30.13	50.06 ± 31.39	0.009 **
Lyso.PC.a.C18.1	10.87 ± 4.65	6.46 ± 5.03	<0.001 ***
Lyso.PC.a.C20.3	0.41[0.29]	0.30 ± 0.25	0.046 *
ΣLyso.PC.a	187.72 ± 66.70	121.02 ± 81.21	<0.001 ***
ΣLyso.PC	192.48 ± 68.15	123.89 ± 83.11	<0.001 ***
**Phosphatidylcholines:**			
PC.aa.C30.0	1.41[0.73]	1.06[1.19]	0.046 *
PC.aa.C32.0	9.21[2.93]	7.97[4.52]	0.03 *
PC.aa.C34.1	103.78 ± 52.01	74.63 ± 49.20	0.01 *
PC.aa.C36.1	28.69 ± 13.61	20.08 ± 13.86	0.006 **
PC.ae.C30.0	0.13[0.10]	0.08[0.15]	0.03 *
PC.ae.C34.0	0.99[0.35]	0.63[0.56]	0.003 **
PC.ae.C36.0	0.29 ± 0.13	0.18 ± 0.12	<0.001 ***
PC.ae.C36.1	2.14[1.07]	1.43[1.20]	0.006 **
PC.ae.C38.3	2.64 ± 1.22	1.92 ± 1.23	0.01 *
PC.ae.C40.2	0.77 ± 0.44	0.51 ± 0.32	0.003 **
**Sphingomyelins:**			
SM.a.C30.1	0.32 [0.23]	[0.17]0.18	0.01
SM.a.C31.1	0.23 ± 0.13	0.15 ± 0.10	0.005 **
SM.a.C32.2	0.65 ± 0.27	0.47 ± 0.27	0.004 **
SM.a.C33.2	0.17 ± 0.09	0.12 ± 0.07	0.007 **
SM.a.C35.1	2.80 ± 1.00	2.09 ± 1.13	0.003 **
SM.a.C35.2	0.36 ± 0.17	0.27 ± 0.15	0.01 *
SM.a.C36.1	17.46 ± 5.64	13.48 ± 7.42	0.008 **
**Acyl carnitines:**			
Carn.a.C8.1	0.06[0.05]	0.09[0.09]	0.01 *

Data for individual fatty acids and fatty acid classes are given as percentage (FA%) of total fatty acids. Polar lipids and polar lipid classes are expressed as μmol/L. Normally distributed values are presented as mean ± SD, and significance assessed by two-tail paired Student’s *t*-test. Non-normal distributed variables are presented as median [interquantile range, IQR], and significance assessed by Wilcoxon matched paired signed-rank test. *p*-values are expressed as *** *p* <**0.001, ** *p* < 0.01, * *p* < 0.05, ns non-significant. AA arachidonic acid; DHA docosahexaenoic acid; LA linoleic acid; Lyso.PC.a acyl-lysophosphatidylcholines; PC.aa diacyl-phosphatidylcholines; PCae acyl-alkyl-phosphatidylcholines; PUFA polyunsaturated fatty acids; SM sphingomyelins.

**Table 5 nutrients-12-02163-t005:** Significant results from the linear models for the change in (**A**) total glycerophospholipid fatty acids and (**B**) polar lipid species with child age at baseline in severe malnourished children treated with RUTF.

Metabolite	Regression Coefficient (β)	*p*-Value	*p*-Value (after FDR Correction)
**A. Total Glycerophospholipid Fatty Acids**			
**Saturated fatty acids:**			
C14:0	−0.06	0.002	0.013 *
C18:0	0.07	0.001	0.010 *
ΣSFA	0.07	0.001	0.010 *
**Monounsaturated fatty acids:**			
C15:1	−0.08	0.001	0.010 *
C16:1n-7	−0.07	0.010	0.045 *
C18:1n-9	−0.08	0.001	0.010 *
C22:1t	−0.07	0.011	0.045 *
ΣMUFA	−0.08	0.001	0.010 *
**Polyunsaturated fatty acids:**			
C20:3n.6	0.06	0.007	0.037 *
**B. Polar lipid species**			
**Lysophosphatidylcholines:**			
lyso.PC.a.C16.0	0.06	0.014	0.030 *
lyso.PC.a.C18.0	0.05	0.012	0.027 *
lyso.PC.a.C18.1	0.06	0.016	0.034 *
lyso.PC.a.C18.2	0.08	0.005	0.024 *
lyso.PC.a.C20.3	0.08	0.001	0.015 *
lyso.PC.a.C20.5	0.06	0.010	0.026 *
ΣLyso.PC.a	0.06	0.008	0.026 *
ΣLyso.PC	0.06	0.009	0.026 *
**Phosphatidylcholines:**			
PC.aa.C32.0	0.04	0.011	0.026 *
PC.aa.C34.1	0.06	0.010	0.026 *
PC.aa.C34.2	0.06	0.008	0.026 *
PC.aa.C34.3	0.05	0.008	0.026 *
PC.aa.C36.1	0.05	0.020	0.041 *
PC.aa.C36.2	0.07	0.004	0.024 *
PC.aa.C36.3	0.07	0.001	0.015 *
PC.aa.C36.4	0.06	0.009	0.026 *
PC.aa.C36.5	0.07	0.003	0.021 *
PC.aa.C38.2	0.07	0.002	0.016 *
PC.aa.C38.3	0.07	0.001	0.015 *
PC.aa.C38.4	0.07	0.004	0.024 *
PC.aa.C38.5	0.08	0.001	0.015 *
PC.aa.C38.6	0.06	0.010	0.026 *
PC.aa.C40.4	0.08	0.001	0.015 *
PC.aa.C40.5	0.08	<0.001	0.015 *
PC.aa.C40.6	0.06	0.005	0.024 *
PC.aa.C44.12	0.07	0.002	0.018 *
ΣPC.aa	0.07	0.003	0.021 *
PC.ae.C32.0	0.04	0.010	0.026 *
PC.ae.C32.1	0.05	0.005	0.024 *
PC.ae.C32.2	0.05	0.014	0.030 *
PC.ae.C34.1	0.04	0.008	0.026 *
PC.ae.C34.2	0.07	<0.001	0.015 *
PC.ae.C34.3	0.08	0.001	0.015 *
PC.ae.C34.4	0.07	<0.001	0.015 *
PC.ae.C36.1	0.03	0.026	0.049
PC.ae.C36.2	0.05	0.010	0.026 *
PC.ae.C36.3	0.07	0.001	0.015 *
PC.ae.C36.4	0.06	0.010	0.026 *
PC.ae.C36.5	0.06	0.012	0.027 *
PC.ae.C36.6	0.05	0.023	0.044 *
PC.ae.C38.3	0.06	0.005	0.024 *
PC.ae.C38.4	0.06	0.010	0.026 *
PC.ae.C38.5	0.06	0.014	0.030 *
PC.ae.C38.6	0.06	0.011	0.026 *
PC.ae.C40.0	0.07	0.001	0.015 *
PC.ae.C40.1	0.06	0.014	0.030 *
PC.ae.C40.2	0.05	0.005	0.025 *
PC.ae.C40.4	0.07	0.007	0.026 *
PC.ae.C40.5	0.08	0.002	0.016 *
PC.ae.C42.6	0.07	0.008	0.026 *
ΣPC.ae	0.07	0.003	0.021 *
ΣPC	0.07	0.003	0.021 *
**Sphingomyelins:**			
SM.a.C34.1	0.05	0.023	0.044 *
SM.a.C34.2	0.07	0.006	0.025 *
SM.a.C36.2	0.06	0.011	0.026 *
SM.a.C36.3	0.06	0.021	0.042 *
SM.a.C38.1	0.06	0.010	0.026 *
SM.a.C38.2	0.06	0.009	0.026 *
SM.a.C40.2	0.07	0.007	0.026 *
SM.a.C42.3	0.06	0.009	0.026 *
SM.a.C42.4	0.08	0.001	0.015 *
SM.a.C42.2	0.06	0.021	0.042 *
ΣSM	0.06	0.021	0.042 *

*p*-values are expressed as * *p *< 0.05. Lyso.PC.a acyl-lysophosphatidylcholines; MUFA monounsaturated fatty acids; PC.aa diacyl-phosphatidylcholines; PCae acyl-alkyl-phosphatidylcholines; PUFA polyunsaturated fatty acids; SFA saturated fatty acids; SM sphingomyelins.

**Table 6 nutrients-12-02163-t006:** Significant results from the linear models for the change in (**A**) total glycerophospholipid fatty acids and (**B**) phospholipid species with child’s breastfeeding status at baseline in severe malnourished children treated with RUTF.

Metabolite	Regression Coefficient (β)	*p*-Value	*p*-Value (after FDR Correction)
**A. Total Glycerophospholipid Fatty Acids**			
**Saturated fatty acids:**			
C18:0	1.00	0.002	0.028 *
SFA	0.92	0.004	0.035 *
**Monounsaturated fatty acids:**			
C18.1n-9	−1.24	0.001	0.028 *
MUFA	−1.22	0.002	0.028 *
C18:3n−3	−1.19	0.006	0.045 *
**B. Polar lipid species**			
**Lysophosphatidylcholines:**			
lyso.PC.a.C18.0	0.66	0.017	0.043 *
**Phosphatidylcholines:**			
PC.aa.C30.0	0.80	0.003	0.012 *
PC.aa.C32.0	0.68	0.006	0.018 *
PC.aa.C32.2	0.71	0.011	0.028 *
PC.aa.C34.2	1.13	0.001	0.006 **
PC.aa.C34.4	0.92	0.007	0.020 *
PC.aa.C34.5	0.95	0.014	0.035 *
PC.aa.C36.2	1.10	0.001	0.007 **
PC.aa.C36.3	1.15	0.001	0.006 **
PC.aa.C36.4	1.30	0.000	0.006 **
PC.aa.C36.5	1.05	0.001	0.007 **
PC.aa.C38.2	0.89	0.007	0.020 *
PC.aa.C38.3	1.08	0.001	0.006 **
PC.aa.C38.4	1.26	0.001	0.006 **
PC.aa.C38.5	1.21	0.000	0.006 **
PC.aa.C38.6	1.19	0.001	0.006 **
PC.aa.C40.4	1.21	0.000	0.006 **
PC.aa.C40.5	1.24	0.000	0.006 **
PC.aa.C40.6	1.09	0.001	0.006 **
PC.aa.C44.12	0.93	0.005	0.018 *
ΣPC.aa	1.11	0.001	0.006 **
PC.ae.C32.0	0.53	0.022	0.050
PC.ae.C32.1	0.75	0.005	0.017 *
PC.ae.C32.2	0.76	0.018	0.043 *
PC.ae.C34.2	0.85	0.007	0.020 *
PC.ae.C34.3	0.91	0.010	0.027 *
PC.ae.C34.4	0.99	0.000	0.006 **
PC.ae.C36.2	0.66	0.021	0.049 *
PC.ae.C36.3	0.83	0.013	0.034 *
PC.ae.C36.4	1.07	0.003	0.012 *
PC.ae.C36.5	1.11	0.003	0.011 *
PC.ae.C36.6	1.16	0.000	0.006 **
PC.ae.C38.0	1.02	0.004	0.015 *
PC.ae.C38.3	1.00	0.001	0.007 **
PC.ae.C38.4	1.09	0.003	0.012 *
PC.ae.C38.5	1.15	0.001	0.007 **
PC.ae.C38.6	1.19	0.001	0.007 **
PC.ae.C40.0	1.16	0.000	0.006 **
PC.ae.C40.2	0.64	0.012	0.032 *
PC.ae.C40.4	1.29	0.001	0.007 **
PC.ae.C40.5	1.33	0.000	0.006 **
PC.ae.C40.6	1.10	0.004	0.015 *
PC.ae.C42.6	1.11	0.002	0.008 **
ΣPC.ae	1.04	0.001	0.007 **
ΣPC	1.11	0.001	0.006 **
**Sphingomyelins:**			
SM.a.C32.1	0.93	0.009	0.025 *
SM.a.C32.2	0.86	0.006	0.018 *
SM.a.C38.1	1.00	0.005	0.017 *
SM.a.C38.2	0.99	0.005	0.017 *
SM.a.C40.2	0.83	0.022	0.049 *
SM.a.C42.3	0.77	0.022	0.049 *
SM.a.C42.4	1.14	0.001	0.006 **
Carn.a.C14.0	1.08	0.004	0.014 *

*p*-values are expressed as ** *p* < 0.01, * *p* < 0.05. Lyso.PC.a acyl-lysophosphatidylcholines; MUFA monounsaturated fatty acids; PC.aa diacyl-phosphatidylcholines; PCae acyl-alkyl-phosphatidylcholines; PUFA polyunsaturated fatty acids; SFA saturated fatty acids; SM sphingomyelins.

**Table 7 nutrients-12-02163-t007:** Significant results from the linear models for the change in (**A**) total glycerophospholipid fatty acids and (**B**) phospholipid species with hemoglobin value at baseline in severe malnourished children treated with RUTF.

Metabolite	Regression Coefficient (β)	*p*-Value	*p*-Value (after FDR Correction)
**A. Fatty acids**			
**Saturated fatty acids:**			
C17:0	−0.22	0.001	0.009 **
**Polyunsaturated fatty acids:**			
C18:2*tt*	−0.27	<0.001	0.009 **
**B. Phospholipid species**			
**Phosphatidylcholines:**			
PC.aa.C30.0	−0.18	0.001	0.010 *
PC.aa.C32.0	−0.12	0.017	0.034 *
PC.aa.C32.1	−0.12	0.007	0.022 *
PC.aa.C32.2	−0.21	0.000	0.010 *
PC.aa.C32.3	−0.18	0.001	0.010 *
PC.aa.C34.3	−0.15	0.005	0.022 *
PC.aa.C34.4	−0.22	0.001	0.010 *
PC.aa.C36.1	−0.15	0.012	0.027 *
PC.aa.C36.3	−0.18	0.006	0.022 *
PC.aa.C36.4	−0.21	0.002	0.013 *
PC.aa.C36.5	−0.21	0.001	0.010 *
PC.aa.C38.3	−0.20	0.001	0.010 *
PC.aa.C38.4	−0.22	0.002	0.011 *
PC.aa.C38.5	−0.22	0.001	0.010 *
PC.aa.C38.6	−0.20	0.004	0.017 *
PC.aa.C40.4	−0.22	0.001	0.010 *
PC.aa.C40.5	−0.27	0.000	0.004 **
PC.aa.C40.6	−0.20	0.001	0.010 *
PC.aa.C44.12	−0.18	0.007	0.022 *
ΣPC.aa	−0.17	0.008	0.023 *
PC.ae.C30.0	−0.16	0.003	0.017 *
PC.ae.C32.0	−0.12	0.008	0.022 *
PC.ae.C32.1	−0.14	0.009	0.024 *
PC.ae.C34.0	−0.14	0.005	0.020 *
PC.ae.C34.1	−0.13	0.007	0.022 *
PC.ae.C36.1	−0.14	0.002	0.010 *
PC.ae.C36.2	−0.14	0.014	0.029 *
PC.ae.C36.3	−0.17	0.010	0.025 *
PC.ae.C36.4	−0.18	0.011	0.027 *
PC.ae.C36.5	−0.18	0.012	0.027 *
PC.ae.C38.3	−0.18	0.002	0.012 *
PC.ae.C38.4	−0.21	0.003	0.017 *
PC.ae.C38.5	−0.21	0.004	0.018 *
PC.ae.C38.6	−0.20	0.006	0.022 *
PC.ae.C40.0	−0.27	0.000	0.005 **
PC.ae.C40.1	−0.15	0.017	0.034 *
PC.ae.C40.2	−0.13	0.009	0.024 *
PC.ae.C40.4	−0.20	0.010	0.025 *
PC.ae.C40.5	−0.21	0.003	0.017 *
PC.ae.C40.6	−0.19	0.011	0.026 *
PC.ae.C42.6	−0.24	0.001	0.010 *
ΣPC.ae	−0.19	0.004	0.017 *
ΣPC	−0.18	0.007	0.022 *
SM.a.C30.1	−0.13	0.007	0.022 *
SM.a.C31.1	−0.15	0.001	0.010 *
SM.a.C32.1	−0.17	0.007	0.022 *
SM.a.C32.2	−0.18	0.001	0.010 *
SM.a.C33.1	−0.13	0.024	0.045 *
SM.a.C33.2	−0.12	0.008	0.023 *
SM.a.C35.1	−0.17	0.006	0.022 *
SM.a.C35.2	−0.17	0.001	0.010 *
SM.a.C36.1	−0.18	0.012	0.027 *
SM.a.C36.2	−0.20	0.006	0.022 *
SM.a.C38.1	−0.19	0.009	0.023 *
SM.a.C38.2	−0.18	0.019	0.037 *
SM.a.C39.1	−0.15	0.017	0.034 *
SM.a.C41.2	−0.18	0.012	0.027 *
SM.a.C42.2	−0.18	0.014	0.029 *
SM.a.C42.3	−0.22	0.002	0.010 *
SM.a.C42.4	−0.24	0.000	0.010 *
SM.a.C42.6	−0.18	0.010	0.025 *
**Acyl carnitines:**			
Carn.a.C9.0	−0.22	0.007	0.022 *

*p*-values are expressed as ** *p* < 0.01, * *p* < 0.05. Lyso.PC.a acyl-lysophosphatidylcholines; PC.aa diacyl-phosphatidylcholines; PCae acyl-alkyl-phosphatidylcholines; PUFA polyunsaturated fatty acids; SM sphingomyelins; *t* trans.

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
