# Peer review of "Impact of Treatment with RUTF on Plasma Lipid Profiles of Severely Malnourished Pakistani Children"

_nutrients, 2020, doi:10.3390/nu12072163_

Round 1

Reviewer 1 Report

The authors have done an extensive analysis of lipid species in malnourished children at baseline and 3 months later

General comments:

  1. Many previous studies had shown depletion of EFA and increased levels of non-essential FA and MUFA in malnutrition, decreased conversion of EFA to long chain PUFA, and a decrease in saturated FA after RUTF treatment. What was the purpose of this more detailed analysis? What was the underlying hypothesis?
  2. There are no data on baseline food intake or composition; thus the study provides no new information regarding the cause of the baseline EFA depletion.
  3. The authors studied the children at baseline and infer that the findings at 3 mo represent the effects of RUTF.  This may be not be the case, as: (a) the children could have eaten foods other than RUTF; (b) the amount of breast milk consumed relative to RUTF is not assessed; (c) the clinical status of the children (infection, etc) might have changed during the 3 months, etc. 
  4. The physiological significance of the magnitude of the changes in lipid species is never addressed but changes in lipid species are not clearly related to clinical status (auxologic, cognitive, etc)
  5. The authors state that breastfeeding supports higher PUFA but this was not tested directly

Author Response

Response to the Reviewers ‘Comments

We would like to express our sincere thanks to the reviewers and editors for the corrections and the valuable suggestions to improve the quality of the manuscript. As per the comments provided, we have revised the manuscript and uploaded a copy of the revised version with the tracked changes highlighted in yellow. In the authors’ response to reviewers’ comments the changes are highlighted in yellow.

Listed below the reviewers’ comments and their responses:

On the Reviewer #1’ Comments:

General comments:

Comment (1): Many previous studies had shown depletion of EFA and increased levels of non-essential FA and MUFA in malnutrition, decreased conversion of EFA to long chain PUFA, and a decrease in saturated FA after RUTF treatment. What was the purpose of this more detailed analysis? What was the underlying hypothesis?

Response:

The authors would like to thank the reviewer for this important comment which would allow us to emphasize the novelty of this research work. It is known that the balance between saturated (SFA), monounsaturated (MUFA) and polyunsaturated fatty acids (PUFA) within phospholipids (PL) plays a role in sustaining the optimal biophysical properties of cellular membranes (Antonny et al. 2015). Thus, perturbation in this balance due to malnutrition as well as changes in response to RUTF treatment would in either case impact the crucial cellular processes (i.e. trafficking, function of membrane-anchored proteins, etc.).

It has been demonstrated in other studies that fatty acids (FA) from the diet can redistribute within PL in a very selective manner, with phosphatidylcholine (PC) being the preferred sink for this redistribution (Bacle et al. 2020). Thus, investigating the fatty acid redistribution within PL in response to a dietary treatment “RUTF” is important because:

  • Most of the malnutrition studies focused on the metabolic or lipid changes observed in malnourished in comparison to heathy children rather than those in response to nutritional intervention in malnourished children (Bartz et al. 2014; Di Giovanni et al. 2016)
  • Additionally, lipidomics methods applied in previous studies typically report only sum FA compositions of lipid species. Therefore, applying a lipidomics approach separating individual glycerophospholipids and sphingolipids, and carnitines (free carnitine and acyl carnitines) in addition to the classical total GPL fatty acids would assign additional detail to the measured species allowing to better visualize the involved metabolic pathways producing the fatty acyl constituents, the lipid class and the pathways involved in lipid metabolism and turnover.

Our aim was to monitor the responses of the PL species bearing polyunsaturated FA chains, and others with two saturated fatty acyl chains (e.g. PC X:0) in response to RUTF treatment to provide an understanding of the impact of RUTF on fatty acid rearrangements in PL. In other words, we aimed to monitor the selective selection of FA for PL synthesis in response to RUTF treatment.

In the literature, severe malnutrition, in particular kwashiorkor, has been linked with changes in hepatic metabolic function (Bhutta et al. 2017), especially hepatic lipid oxidation which might explain the occurrence of hepatic steatosis (Badaloo et al. 2006; Williams 1933). Therefore, changes in PL species composition in malnourished children in response to RUTF treatment would not be surprising, because the liver plays a key role in lipid metabolism being the hub of FA synthesis and lipid circulation through lipoprotein synthesis (Nguyen et al. 2008).

References:

Antonny B, Vanni S, Shindou H, Ferreira T. From zero to six double bonds: phospholipid unsaturation and organelle function. Trends Cell Biol. 2015;25(7):427-436. doi:10.1016/j.tcb.2015.03.004

Bacle A, Kadri L, Khoury S, et al. A comprehensive study of phospholipid fatty acid rearrangements in metabolic syndrome: correlations with organ dysfunction. Dis Model Mech. 2020;13(6): dmm043927. Published 2020 Jun 15. doi:10.1242/dmm.043927

Badaloo AV, Forrester T, Reid M, Jahoor F. Lipid kinetic differences between children with kwashiorkor and those with marasmus. Am J Clin Nutr. 2006; 83: 1283–1288.

Bartz S, Mody A, Hornik C, et al. Severe acute malnutrition in childhood: hormonal and metabolic status at presentation, response to treatment, and predictors of mortality. J Clin Endocrinol Metab. 2014; 99(6): 2128–2137. doi:10.1210/jc.2013-4018

Bhutta ZA, Berkley JA, Bandsma RHJ, et al. Severe childhood malnutrition. Nat Rev Dis. Primers. 2017 Sep; 3: 17067. DOI: 10.1038/nrdp.2017.67.

Di Giovanni V, Bourdon C, Wang DX, Seshadri S, Senga E, Versloot CJ, et al. Metabolomic Changes in Serum of Children with Different Clinical Diagnoses of Malnutrition. J Nutr 2016; 146(12): 2436–44. doi: 10.3945/jn.116.239145.

Nguyen P, Leray V, Diez M, et al. Liver lipid metabolism. J Anim Physiol An N 2008, 92: 272-283. doi:10.1111/j.1439-0396.2007.00752.x

Williams CD. A nutritional disease of childhood associated with a maize diet. Arch Dis Child. 1933; 8: 423–433.

We have updated the introduction to further clarify the usefulness of investigating the individual PL species in addition to the classical total GPL-FA analyses, as shown in lines 67-81 in the revised version of the manuscript with tracked changes to be as follows:

“Most of the previous studies on malnutrition focused on the metabolic or lipid changes observed in malnourished in comparison to heathy children rather than those in response to nutritional intervention in malnourished children [11, 12]. Also, lipidomics methods applied in previous studies typically report only sum FA compositions of lipid species. In the present study, we examined the both the glycerophospholipid (GPL) profiles and the comprising molecular lipid species in children with SAM before and 3 months after treatment with RUTF. It is known that the balance between saturated (SFA), monounsaturated (MUFA) and polyunsaturated fatty acids (PUFA) within phospholipids (PL) plays a role in sustaining the optimal biophysical properties of cellular membranes [13]. Thus, perturbation in this balance due to malnutrition as well as changes in response to RUTF treatment would in either case impact the crucial cellular processes (i.e. trafficking, function of membrane-anchored proteins, etc.) [13]. It has been demonstrated in other studies that fatty acids (FA) from the diet can redistribute within PL in a very selective manner, with phosphatidylcholine (PC) being the preferred sink for this redistribution [14]. Thus, investigating the fatty acid redistribution within PL in response to a dietary treatment “RUTF” would be advantageous to better visualize the involved metabolic pathways producing the fatty acyl constituents, the lipid classes and the pathways involved in lipid metabolism and turnover.  We also studied the impact of different factors potentially associated with metabolic changes, including child age at enrollment, anthropometry, sex, breastfeeding, and hemoglobin levels.”

Comment (2): There are no data on baseline food intake or composition; thus the study provides no new information regarding the cause of the baseline EFA depletion.

Response:

  • We would like to thank the reviewer for this important comment. Baseline data are available for both the food intakes based on a 24-hour dietary recall (24 h). Accordingly, in the revised version of the manuscript with tracked changes, a paragraph was added to describe the dietary data shown in p.5, lines 125-131, as follows:

“Dietary data including breastfeeding were recorded based on a 24-hour dietary recall (24 h) for the food consumed by the children in the previous 24 h. The food group lists were defined based on a questionnaire from The World Health Organization (WHO) [17] and were aggregated into 7 groups: 1) cereals, roots, and tubers; 2) legumes and nuts; 3) dairy products (milk, yogurt, cheese); 4) flesh foods (meat, fish, poultry, and liver/organ meats); 5) eggs; 6) vitamin A rich fruits and Vegetables; 7) other fruits and vegetables. For breastfeeding, information was provided on whether the child was ever/currently and if breastfed, with what frequency over the previous 24 h. “

  • Information on the study participants (%) by food group both at baseline and after treatment with RUTF for 3 months are provided in Table 3 in the revised version of the manuscript with tracked changes.

Table 3. Study participants (%) by food group at baseline and after treatment with RUTF

Food Groups

Dietary Components

At baseline

(%)

After treatment

(%)

p-value

1

Grains, roots and tubers

100

100[7]

ns

2

Legumes and Nuts

12.2

20.6[7]

ns

3

Dairy Products (milk, yogurt, cheese)

51.2

55.9[7]

ns

4

Flesh foods (meat, fish, poultry, and liver/organ meats)

4.9

11.8[7]

ns

5

Eggs

2.4

5.9[7]

ns

6

Vitamin A rich fruits and Vegetables

12.2

14.7[7]

ns

7

Other fruits and Vegetables

14.6

11.8[7]

ns

8

Breastmilk consumption

5.4±1.6

5.2±1.1[7]

ns

Data are expressed as percentage of total study participants by group according to 24 h recall rate. Breastmilk consumption is expressed as frequency of breastfeeds according to 24 h recall and is described by mean±SD. Paired differences between the child’s characteristics at baseline and after treatment with RUTF were evaluated via McNemar's Chi-squared and Chi-squared tests for categorical and continuous variables, respectively. P-values expressed as ns non-significant. Numbers in square brackets indicate the numbers of missing observations.

Based on the data, at baseline, all the children consumed a diet based on grains, roots and tubers with only 4.9%, 12.2% and 14.6% of them consuming other flesh foods, vitamin A rich fruits and vegetables, and other fruits and vegetables, respectively. This section has been added in the revised version of the manuscript with tracked changes as shown in p. 8, lines 222-234.

Based on this data, we suggest several hypotheses behind EFA in malnourished children at baseline:

  • The presented data matched with the consensus that diets in the majority of low-income countries are based mainly on stable foods (cereals, legumes and roots) which generally have a low content of PUFA, especially n-3 PUFA. This dietary pattern might barely meet the recommendations for n-6 PUFA but not n-3 PUFA or n-6/n-3 ratio. For Pakistani households, cereals are the main source of calories (60%) followed by 12% from oils, and 10% from sugars (World Bank 2016).
  • Apart from this, the energy density of a diet based on cereals is low which means that the food is too bulky, and the child will not be able to eat adequate amounts. The most important factor affecting the energy density is the fat content (energy density 9 kcal/g) which is more than double that of protein and carbohydrate (4 kcal/g). Infants and young children have a limited gastric capacity and an energy requirement (/unit body weight) about three times as high as adults. Therefore, even non-malnourished children given a low-energy density diet, may not be able to eat adequate amounts because of the bulkiness of the diet (Michaelsen et al. 2009).
  • Dietary fat is not only important for the energy requirement, but also plays a role as a source of essential fatty acids (EFA) and in allowing adequate absorption of fat-soluble vitamins. Therefore, we assume the EFA depletion issue to be the most relevant to malnutrition (Michaelsen et al. 2009).
  • The lack of food diversity in the diets in low-income countries could be one of the reasons behind the deficiency in PUFA (Shabnam et al. 2016) which in most diets are provided by vegetable oils (EFA) or from meat and fish (LC-PUFA) which are not adequately provided in a diet mainly based on cereals (evident in the present population)
  • Infants and young children in low-income countries are more prone to low birthweights and thus poor fetal stores, which makes them vulnerable and dependent on a postnatal dietary supply of n-3 LCPUFA.
  • Children in low-income countries suffer environmental stress (e.g. infections) which increases their requirement for PUFA (Calder 2010; Katona and Katona-Apte 2008; Rodríguez et al. 2011; Tomkins and Watson 1989; Yaméogo et al. 2017) (this hypothesis is not strongly supported in our findings as explained in detail in the authors’ response to reviewer’s comment 3.

The findings and their explanation in the discussion as seen in the revised version of the manuscript with tracked changes (p.18: lines 368-391, as follows:

“The dietary pattern could be one of the reasons underlying the low EFA supply in those children at baseline. This hypothesis is supported by the dietary data of the children in this study (Table 3) demonstrating that all the children consumed a diet based on grains with only a low percentage of children consuming flesh foods, vitamin A rich fruits and vegetables, or other vegetables or fruits. This diet characterized by a lack of food diversity does not likely or perhaps barely meet the recommendations for n-6 PUFA, but not n-3 PUFA or n-6/n-3 ratio. The presented data matched with the consensus that diets in most of the low-income countries are based mainly on stable foods (cereals, legumes and roots) which generally have a low content of PUFA, especially n-3 PUFA [31]. For Pakistani households, cereals are the main source of calories (60%) followed by 12% from oils, and 10% from sugars [32]. Apart from this, the energy density of a diet based on cereals is low which means that the food is too bulky, and the child will not be able to eat adequate amounts. Infants and young children have a limited gastric capacity and an energy requirement (/unit body weight) about three times as high as adults. Therefore, even non-malnourished children given a low-energy density diet, may not be able to eat adequate amounts because of the bulkiness of the diet [33]. 

Infants/young children in low-income countries could be also more prone to low birthweights and thus poor fetal stores, which could make them vulnerable and dependent on a postnatal dietary supply of LCPUFA which if not sufficiently provided in diet would lead to EFA depletion [33]. Another hypothesis involves exposure of children in poor or low-income countries to environmental stress (e.g. infections) which increases their requirement for PUFA, as previously reported [26, 33-37]. To test this hypothesis,  we investigated the associations between baseline co-morbidity and GPL-FA which showed no significant associations with the GPL-FA, at baseline after adjustment for age and sex, except for C20:3n-6 and C18:3n-6 whose values tended to be lower in ill children at baseline (not significant after FDR correction) (Supplementary Table 9). Therefore, the present findings do not sufficiently support the impact of the infection state on the GPL-FA profiles including PUFA..”

References:

Calder PC. Omega-3 fatty acids and inflammatory processes. Nutrients. 2010; 2: 355–374. doi: 10.3390/nu2030355.

Katona P, Katona-Apte J. The interaction between nutrition and infection. Clin Infect Dis. 2008; 46: 1582–1588. doi: 10.1086/587658.

Michaelsen KF, Hoppe C, Roos N, et al. Choice of foods and ingredients for moderately malnourished children 6 months to 5 years of age. Food Nutr Bull. 2009; 30(3 Suppl): S343-S404. doi:10.1177/15648265090303S303.

Rodríguez L, Cervantes E, Ortiz R. Malnutrition and gastrointestinal and respiratory infections in children: a public health problem. Int J Environ Res Publ Health. 2011; 8: 1174–1205. doi: 10.3390/ijerph8041174.

Shabnam N, Santeramo FS, Asghar Z, and Seccia A. The Impact of the Food Price Crises on the Demand for Nutrients in Pakistan. J South Asian Dev 2016; 11(3): 305–327.

Tomkins A, Watson F. Malnutrition and Infection - A Review – Nutrition Policy Discussion Paper No. 5. United Nations - Administrative Commitee on Coordination - Subcommitee on Nutrition; 1989: 3-30.

World Bank 2016. Revisiting the Poverty Debate in Pakistan: Forensics and the Way Forward. Washington, DC: World Bank.

Yaméogo CW, Cichon B, Fabiansen C, Rytter MJH, Faurholt-Jepsen D, Stark KD, et al. Correlates of whole-blood polyunsaturated fatty acids among young children with moderate acute malnutrition. Nutr J. 2017; 16: 44.

Comment 3. The authors studied the children at baseline and infer that the findings at 3 mo represent the effects of RUTF.  This may be not be the case, as: (a) the children could have eaten foods other than RUTF; (b) the amount of breast milk consumed relative to RUTF is not assessed; (c) the clinical status of the children (infection, etc) might have changed during the 3 months, etc.?

Response: We would like to thank the reviewer for this important comment. Within the study, data on the food intakes, frequency of breastfeeding and clinical status were collected at baseline and at follow up (3 months of RUTF treatment), using questionnaires and clinical examination.

  • Co-morbidity was used as a proxy for the clinical status of the enrolled children and was evaluated:

(a) at baseline: based on whether the child was suffering diarrhea and/or, repeated episodes of cough/flu/sore throat/difficulty in breathing, and/or pneumonia, and/or measles.

(b) at follow up: based on whether the child was suffering from diarrhea, and/or vomiting, and/or flu/cough, fast breathing /difficulty in breathing, and/or fever either on the day of the examination or during the last fourteen days, prior to the examination day.

The evaluation of the clinical status using co-morbidity was described in the revised version of the manuscript with tracked changes (p. 4, lines 113-116), as follows:

“Additionally, data on co-morbidities was defined as a child suffering from (1) At baseline: diarrhea and/or repeated episodes of cough/flu/sore throat and/or pneumonia and/or measles; (2) after treatment:  current/14-day retrospective diarrhea and/or cough/flu/sore throat and/or fever and/or vomiting.”

Also, in the revised version of the manuscript, information on the clinical status was added to the characteristics of the population included in the study at baseline and after treatment with RUTF) (Table 2).

  • Food intakes are determined based on consumption of the study participants to certain food groups according to a 24-hour dietary recall (24 h) both at baseline and follow up (Table 3 in the revised version of the manuscript)
  • Frequency of breastfeeding is determined both at baseline and after treatment based on a 24-hour dietary recall. (Table 3 in the revised version of the manuscript)

Accordingly:

  • Paired differences between the co-morbid status at baseline and after treatment were ca-lculated using McNemar's Chi-squared test and the results showed no significant differe-nces at p-values <0.05. We also investigated the differences in the presences of the two p-arameters for which data is available both at baseline and after treatment (diarrhea & cough/flu/sore throat and there were also no significant differences (Table 2).
  • Additionally, we have investigated the associations between the co-morbidity at baseline and the GPL-FA, ill children were found to show tendency for lower levels of C20.3n-6 and C18.3n-6 (not significant after FDR correction) (Supplementary Table 9). These findings do not sufficiently support the impact of the baseline infection status on the GPL-FA profiles including P-UFA. Concomitantly, there was no significant difference in the clinical status of the children (i.e. infection) at baseline and after treatment. Therefore, we exclude that the detected effects in the GPL-FA profiles, especially PUFA to be a result of the change in the clinical status.

Table 2. Characteristics of the population included in the study at baseline and after treatment with RUTF

Characteristics

At baseline

After treatment

p-value

Sex

                Male (22; 54%), Females (19; 46%)

Breastfeeding status (y/n)

14 (34%)

14 (34%)

ns

Hemoglobin (Hb; (g/dl))

8.35±2.13 [6]

9.71±1.52 [7]

**

Length (cm)

77.14±8.09

78.94± 7.90 [4]

***

Weight (Kg)

7.36±1.45

8.78±1.67 [4]

***

WHZ z-scores

-3.56±0.56

-1.91±0.76 [7]

***

Co-morbidity (y/n)

23 (56%)

18 (53%)

ns

Diarrhea (y/n)

14 (34%)

14 (34%)

ns

Repeated episodes of cough/flu/sore throat (y/n)

19 (46%)

11(27%)

ns

Pneumonia (y/n)

1 (2.4%)

na

Measles (y/n)

0(0%)

na

Fever

na

12 (29%)

Vomiting

na

3 (7.3%)

Tabulated values are expressed in ‘mean±standard deviation (SD)’. Numbers in square brackets indicate the numbers of missing observations. P-values are expressed as ***p<0.001, **p<0.01, *p<0.05, ns non-significant. NA not available. Paired differences between the child’s characteristics at baseline and after treatment with RUTF were evaluated via Mc Nemar's Chi-squared and Chi-squared tests for categorical and continuous variables, respectively. Morbidity is defined as a child suffering from (1) At baseline: diarrhea and/or repeated episodes of cough/flu/sore throat and/or pneumonia and/or measles; (2) after treatment:  diarrhea and/or repeated episodes of cough/flu/sore throat and/or fever and/or vomiting.

  • In spite that there were no data available on the relative intakes of the breastmilk relative to the study formula. However using the data available on the frequency of breastfeeding at baseline and after treatment, we calculated the differences between the frequency of breastfeeding at baseline and after treatment using Chi Squared test and there were no significant differences at p-values <0.05 (Table 3).
  • We also investigated the difference between the consumption of the food groups at baseline and after treatment using McNemar's Chi-squared test and no significant differences were detected for all the food groups.

Table 3. Study participants (%) by food group at baseline and after treatment with RUTF

Food Groups

Dietary Components

At baseline

(%)

After treatment

(%)

p-value

1

Grains, roots and tubers

100

100[7]

ns

2

Legumes and Nuts

12.2

20.6[7]

ns

3

Dairy Products (milk, yogurt, cheese)

51.2

55.9[7]

ns

4

Flesh foods (meat, fish, poultry, and liver/organ meats)

4.9

11.8[7]

ns

5

Eggs

2.4

5.9[7]

ns

6

Vitamin A rich fruits and Vegetables

12.2

14.7[7]

ns

7

Other fruits and Vegetables

14.6

11.8[7]

ns

8

Breastmilk consumption

5.4±1.6

5.2±1.1[7]

ns

Data are expressed as percentage of total study participants by group according to 24 h recall rate. Breastmilk consumption is expressed as frequency of breastfeeds according to 24 h recall. Paired differences between the child’s characteristics at baseline and after treatment with RUTF were evaluated via McNemar's Chi-squared and Chi-squared tests for categorical and continuous variables, respectively. P-values expressed as ns non-significant. Numbers in square brackets indicate the numbers of missing observations. NA not available.

Based on the above findings, we hypothesize that the detected changes in the GPL-FA profiles after treatment is a more likely a result of the consumption of the study formula, however we still cannot exclude that it could be due to a combined result of the consumption of the RUTF formula with other factors such as increase in the relative amounts of breastmilk consumed relative to study formula. The discussion section in the revised version of the manuscript with tracked changes (p. 21, lines 476-487), as follows:

“As shown in the previous findings, remarkable changes were detected in the concentrations of GPL-FA species and their comprising molecular species. These changes may be attributed to the consumption of the RUTF formula however it might also be a result of other factors (change in the clinical status, consumption of other foods, changes in the relative quantities of breast milk consumed relative to the study formula) or a combination of two or more of these factors. By examination of the parameters with available data, we found no significant differences in the comorbidities, consumption of the food groups, frequency of breastfeeding after treatment relative to baseline data, as previously shown in the results section.  In addition, the results showed lack of significant associations between the changes in the GPL-FA profiles and a disease status at baseline. On the other hand, we have no available data on the relative consumption of breastmilk relative to the study formula. Therefore, we interpreted the findings presumably a result of the consumption of the RUTF formula, however a possible contribution of other factors cannot be excluded.”

Comment 5: The physiological significance of the magnitude of the changes in lipid species is never addressed but changes in lipid species are not clearly related to clinical status (auxologic, cognitive, etc).

We agree with the reviewer that evaluating the associations between the changes in lipid species with auxologic and cognitive clinical status as outcomes could give a major contribution in the assessment of the physiological effect of the changes in the GPL-FA profiles. Unfortunately, we did not have this opportunity, that is why we added this as one of the study limitations in the revised version of the manuscript with tracked changes (p.22, lines 557-561), as follows:

” Another study limitation is the lack of data on the clinical auxologic and cognitive effects relevant to the changes in lipid species after RUTF treatment which makes the interpretation of the physiological impacts of changes in lipid species compositions rather difficult”

On the other hand, we used other parameters like gain in WHZ z-scores and hemoglobin (Hb) as proxy measures for the physiological significance of the treatment since all these parameters showed significant improvement after treatment relative to baseline. The results showed no significant associations between the change in any of these parameters with GPL-FA. Additionally, the overall infection status at baseline did not show significant associations with the GPL-FA levels at baseline as explained in the author’s response to comment 3. In conclusion, all the markers of the physiological/clinical status were not conclusive as we were expecting and cannot be interpreted on the basis of the changes in lipid species.

The conclusion section has been updated in the revised version of the manuscript with tracked changes (p.24, lines 571-574), as follows:

“In spite that there was a marked improvement detected in the WHZ z-scores and Hb after RUTF treatment, no evidence was found relating this outcome to the detected changes in the GPL-FA, levels and their comprising molecular species.” 

Comment 6: The authors state that breastfeeding supports higher PUFA but this was not tested directly

Response: We agree with the reviewer that the higher PUFA supply through breastfeeding was not directly tested since we have no available data on the PUFA composition of the breastmilk consumed by the children in this study. Accordingly, we updated the discussion section of revised version of the manuscript with tracked changes to highlight this point and presented the findings as a speculation: as follows (p. 22, lines 502-504):

”Despite the lack of available data on the PUFA composition of the breastmilk consumed by the children in this study, we speculate that the LC-PUFA supply with breastfeeding may explain the non-significant trend to a greater increase in LC-PUFA, such as AA and C20.3n.6, and DHA in breastfed infants after intervention, as compared to non-breastfed infants.”

Nevertheless, we based our hypothesis on the established consensus that breast milk is the ‘gold standard’ for most nutrients (Bernt et al. 1995; Smit et al. 2002). Under optimal maternal nutritional and living conditions, breast milk alone can provide enough energy from fat (50±60% of total energy) and all long chain polyunsaturated fatty acids (LC-PUFA) and their parent n-6 and n-3 EFA necessary to meet the infant's requirements for normal development, at least until 4±6 months of age (FAO=WHO 1994; Rocquelin et al. 1998).

Previous studies have reported lower levels of LC-PUFA in formula fed babies in comparison to breast-fed infants (Decsi et al. 1994; Huisman et al. 1996). In other studies, close associations have been observed between the breast-milk fatty acid composition and infant plasma, red blood cells (RBCs) and tissue fatty acids (FA) levels (Andersen et al. 2015; Jensen et al. 2000; Mellies et al. 1979; Pugo-Gunsam et al. 1999). In another study, a positive correlation was found between n-6/n-3 LCPUFA ratio in breast milk and infant RBCs (Much et al. 2013).

According to our findings, breastfeeding was negatively associated with the increase in ALA (Table 5), a positive association was found between breastfeeding with Σn-6 PUFA and ΣPUFA (not significant after FDR correction). We hypothesized that in spite that breast milk is recognized as a good source of LC-PUFAs, it is influenced by maternal diet (Read et al. 1965; Smit et al. 2002).

In a study involving breast milk samples collected over 25 years from 11 different countries, breast milk from Pakistani mothers reported highest levels of LC-SFA, 16:0, 20:3n-9/20:4n-6, n-6/n-3 PUFA, 18:2n-6/18:3n-3, n-6/n-3 PUFA and AA/DHA, and the lowest PUFA and LC-PUFA. The detected profile was majorly attributed to the marginal EFA especially n-3 PUFA, due to the low intake of vegetable oil and the very low fish consumption in North Pakistan (Smit et al. 2002).

Based on that we updated the discussion section in the revised version of the manuscript with tracked changes (p. 22, lines 502-518), as follows:

“Despite the lack of available data on the PUFA composition of the breastmilk consumed by the children in this study, we speculate that the LC-PUFA supply with breastfeeding might provide an explanation for the non-significant trend to a greater increase in LC-PUFA, such as AA and C20.3n.6, and DHA in breastfed infants after intervention, as compared to non-breastfed infants. . This speculation is in line with the established consensus that breastmilk is the ‘gold standard’ for most nutrients [62, 63]. Provided the optimal maternal living and nutritional conditions, breastmilk alone provides not only sufficient energy supply (50±60% of total energy) from fat, but also all LC-PUFA and their parent n-6 and n-3 EFA necessary to meet the infant's requirements for normal development, at least up to 4±6 months of age [64, 65]. Previous studies have reported lower levels of LC-PUFA in formula fed babies in comparison to breastfed infants [66, 67]. In other studies, close associations have been observed between the breastmilk fatty acid composition infant plasma, RBCs [68], and tissue FA levels [69-71]. In another study, a positive correlation was found between n-6/n-3 LC-PUFA ratio in breastmilk at 4 months of lactation and infant RBCs at 4 and 12 months of age [72].

On the other hand, the breastmilk composition was proven to be influenced by maternal diet, especially under conditions where uncertainty regarding the appropriate maternal diet are involved [62]. Perhaps, this would explain why the trends towards higher PUFA observed in breastfed infants in this study were much less pronounced for ∑n-3 PUFA. In North Pakistan, low breastmilk n-3 LC-PUFA levels might be expected because of the predominant use of corn oil and ghee which are low in ALA and do not provide DHA, a low intake of green leafy vegetables that serve as a source of ALA, and lack of consumption of n-3 LC-PUFA-rich fish [62, 73]. “

References:

Andersen SB, Hellgren LI, Larsen MK, Verder H, Lauritzen L. Long-Chain Polyunsaturated Fatty Acids in Breast-Milk and Erythrocytes and Neurodevelopmental Outcomes in Danish Late-Preterm Infants. J Preg Child Health 2015; 2:160. doi: 10.4172/2376-127X.1000160

Bernt KM, Walker WA. Human milk as a carrier of biochemical messages. Acta Paediatr Suppl 1999; 430: 27-41.

Decsi T, Koletzko B. Polyunsaturated fatty acids in infant nutrition. Acta Paediatr Suppl 1994; 395: 31-7.

FAO=WHO. Fats and oils in human nutrition. Report of the FAO=WHO expert consultation on fats and oils in human nutrition (1994).

Huisman M, van Beusekom CM, Lanting CI, Nijeboer HJ, Muskiet FAJ, Boersma ER. Triglycerides, fatty acids, sterols, mono- and disaccharides and sugar alcohols in human milk and current types of infant formula milk. Eur J Clin Nutr 1996; 50: 255-60

Jensen CL, Maude M, Anderson RE & Heird WC. Effect of docosahexaenoic acid supplementation of lactating women on the fatty acid composition of breast milk lipids and maternal and infant plasma phospholipids. Am J Clin Nutr 2000; 71: 292S–299S

Mellies MJ, Ishikawa TT, Gartside PS, et al. Effects of varying maternal dietary fatty acids in lactating women and their infants. Am J Clin Nutr 1979; 32: 299–303.

Much D, Brunner S, Vollhardt C, et al. Breast milk fatty acid profile in relation to infant growth and body composition: results from the INFAT study. Pediatr Res. 2013; 74(2): 230-237. doi:10.1038/pr.2013.82

Pugo-Gunsam P, Guesnet P, Subratty AH, Rajcoomar DA, Maurage C & Couet C Fatty acid composition of white adipose tissue and breast milk of Mauritian and French mothers and erythrocyte phospholipids of their full-term breast-fed infants. Br J Nutr 1999; 82: 263–271.

Read WWC, Lutz PG, Tashjian A. Human milk lipids. II. The influence of dietary carbohydrates and fat on the fatty acids of mature milk. Am J Clin Nutr 1965; 17: 180-3

Rocquelin, G., Tapsoba, S., Dop, M. et al. Lipid content and essential fatty acid (EFA) composition of mature Congolese breast milk are influenced by mothers' nutritional status: Impact on infants' EFA supply. Eur J Clin Nutr 1998; 52, 164–171. https://doi.org/10.1038/sj.ejcn.1600529

Smit EN, Martini IA, Mulder H, Boersma ER, Muskiet FA. Estimated biological variation of the mature human milk fatty acid composition. Prostaglandins Leukot Essent Fatty Acids. 2002; 66(5-6): 549-555. doi:10.1054/plef.2002.0398

Reviewer 2 Report

The article is well designed, well-written and presented. The only limitation is the low number of study participants. It would have been better, if there has been a comparison the the data from the study participants with the fatty acid composition of healthy controls. It is important to have such a comparison to see the efficacy of RUTF in this study group, as the data cannot be compared with the reference values from European children mentioned in the article (Glasser et al., 2010), though it has not been compared. I have noticed a few typos/editorial corrections that should be taken care.

Author Response

On the Reviewer #2’ Comments:

We would like to express our sincere thanks to the reviewers and editors for the corrections and the valuable suggestions to improve the quality of the manuscript. As per the comments provided, we have revised the manuscript and uploaded a copy of the revised version with the tracked changes highlighted in yellow. In the authors’ response to reviewers’ comments the changes are highlighted in yellow.

Listed below the reviewers’ comments and their responses:

General comments:

Comment (1): The article is well designed, well-written and presented. The only limitation is the low number of study participants.

Response: We would like to thank the reviewer for the important comment which we highlighted as one of the study limitations in the revised version of the manuscript with tracked changes (p. 23: lines 552-553).

Comment (2): It would have been better, if there has been a comparison the data from the study participants with the fatty acid composition of healthy controls. It is important to have such a comparison to see the efficacy of RUTF in this study group, as the data cannot be compared with the reference values from European children mentioned in the article (Glasser et al., 2010), though it has not been compared.

Response: We agree with the reviewer that the ideal situation would be to compare the lipid species data from malnourished children with data of healthy children from the same population, however practically it would be difficult to get  an ethical review board approval to collect samples from healthy Pakistani children (p.23, lines 552-555, revised version of the manuscript with tracked changes). However, it is worth saying that in a study investigating the plasma phospholipid FA (PL-FA) composition in normal individuals from 15 different populations (4 from USA, 3 from UK, 2 from Canada, one from each of Denmark, Ireland, Norway, Finland, Japan and Zimbabwe), there were remarkably small differences in the concentrations with the exceptions of the long chain n-3 essential fatty acids in the different populations. These findings showed relatives constancies in the PL-FA compositions despite the difference in race (in North American and European samples, the individuals in the were white, while the individuals in the Zimbabwe sample were relatively poor blacks), geographical locations, diets, and life styles (David et al. 1991).

References:

David F. Horrobin, Kelly Ells, Nancy Morse-Fisher & Mehar S. Manku. Fatty Acid Distribution in Plasma Phospholipids in Normal Individuals from Different Geographical Locations, J Nutr Med. 2009; 2(3): 249-258, DOI: 10.3109/13590849109084122.

Comment (3): I have noticed a few typos/editorial corrections that should be taken care.​

Response: We have proofread the manuscript for correction of grammatical errors and typos as per the reviewer’s comment and highlighted the corrected typos in yellow in the revised version of the manuscript with tracked changes.
